# Nitric Oxide Ameliorates Plant Metal Toxicity by Increasing Antioxidant Capacity and Reducing Pb and Cd Translocation

**DOI:** 10.3390/antiox10121981

**Published:** 2021-12-13

**Authors:** Abolghassem Emamverdian, Yulong Ding, James Barker, Farzad Mokhberdoran, Muthusamy Ramakrishnan, Guohua Liu, Yang Li

**Affiliations:** 1Co-Innovation Center for Sustainable Forestry in Southern China, Nanjing Forestry University, Nanjing 210037, China; ylding@vip.163.com (Y.D.); mfarzad649@hotmail.com (F.M.); ramky@njfu.edu.cn (M.R.); 2Bamboo Research Institute, Nanjing Forestry University, Nanjing 210037, China; 3School of Life Sciences, Pharmacy and Chemistry, Kingston University, Kingston-upon-Thames KT1 2EE, UK; j.barker@kingston.ac.uk; 4Department of Mathematical Sciences, Florida Atlantic University, Boca Raton, FL 33431, USA; yangli@fau.edu

**Keywords:** nitric oxide, bamboo species, lead (Pb), cadmium (Cd)

## Abstract

Recently, nitric oxide (NO) has been reported to increase plant resistance to heavy metal stress. In this regard, an in vitro tissue culture experiment was conducted to evaluate the role of the NO donor sodium nitroprusside (SNP) in the alleviation of heavy metal toxicity in a bamboo species (*Arundinaria pygmaea*) under lead (Pb) and cadmium (Cd) toxicity. The treatment included 200 µmol of heavy metals (Pb and Cd) alone and in combination with 200 µM SNP: NO donor, 0.1% Hb, bovine hemoglobin (NO scavenger), and 50 µM L-NAME, N(G)-nitro-L-arginine methyl ester (NO synthase inhibitor) in four replications in comparison to controls. The results demonstrated that the addition of L-NAME and Hb as an NO synthase inhibitor and NO scavenger significantly increased oxidative stress and injured the cell membrane of the bamboo species. The addition of sodium nitroprusside (SNP) for NO synthesis increased antioxidant activity, protein content, photosynthetic properties, plant biomass, and plant growth under heavy metal (Pb and Cd) toxicity. It was concluded that NO can increase plant tolerance for metal toxicity with some key mechanisms, such as increasing antioxidant activities, limiting metal translocation from roots to shoots, and diminishing metal accumulation in the roots, shoots, and stems of bamboo species under heavy metal toxicity (Pb and Cd).

## 1. Introduction

In recent decades, soil contamination caused by anthropogenic activity has become an environmental dilemma [1]. A national survey of soils in China showed that 16% of the farmland soil was acutely polluted [2]. Today, a significant portion of China’s farmland (19.4%) and a large part of China’s forestland (10.0%), especially the bamboo forests, are contaminated with heavy metals. This heavy metal content increases every year and is deemed to be a significant threat to human health [3]. It is estimated that 13.86% of crops produced in China’s farmlands are exposed to heavy metal pollution [2]. Among them, cadmium and lead are the two most common toxic heavy metals in the region [4]. Cadmium (Cd) is the most toxic heavy metal to humans and the environment and accounts for the highest percentage of heavy metal (7%) toxicity in Chinese agricultural soil and urban areas [2,4,5]. Excess Cd in farmland and agricultural soils leads to various hazardous effects on the environment and on plant organs, such as disrupting the balances of soil micronutrients and macronutrients [6,7] and contributing to root elongation [7]. Additionally, by increasing H_2_O_2_ accumulation, Cd causes oxidative stress, which leads to a reduction in the content of photosynthetic pigments and plant growth [8,9]. Lead (Pb) is another hazardous toxic metal that is widespread in Chinese farmland soils [10]. Lead (Pb) causes many disorders in the plant, such as influencing seed germination and the cell division stage in seed germination by affecting related responsible enzymes and reducing photosynthetic properties, as well as inhibiting plant growth [11].

Nitric oxide (NO) is a small molecule with a short half-life that plays a biological role in plant signaling and can act as a mediator for primary messengers [12,13]. Endogenous SNPs (NO donors) are the main form of NO and can play an effective role in the activation of plant biological signaling in multiple plant functions, from seedling to plant growth [14,15]. On the other hand, the activity of NO synthase (NOS) is affected by NO modulators, such as the application of different l-arginine analogs, including l-NAME (as an NOS inhibitor) and Hb (as an NO scavenger) [16,17], which can clearly demonstrate the role of NO in the enhancement of plant tolerance under abiotic stress. Recently, there has been some evidence that NO has the ability to reduce abiotic stress, especially in plants exposed to heavy metals [17,18,19,20]. In addition to the signaling role of NO in the reduction of heavy metal toxicity, NO decreases metalloid accumulation and can activate Reactive oxygen species (ROS) antioxidants in plants under metal stress [20]. NO protects plant cells from oxidative stress by converting H_2_O_2_ to O_2_ and stimulating H_2_O_2_-suppressing enzymes [21]. This protection has been reported in various plant species exposed to cadmium [22], arsenic [23], and copper [24].

Bamboo plants, with 500 species and 70 genera, cover a large part of Chinese forestland (approximately 6 million hectares) [25,26]. Bamboo, with a fast-growing period and high biomass [27], plays a unique role in the detoxification of the environment and phytoremediation [28], and bamboo in southern China acts as a forest resource and economically valued plant that is widely involved in the everyday life of local households [25]. According to the report by the State Forestry Administration of China (2012), the bamboo industry, a growing economy in China, accounts for an excess of $19.7 billion (USD) of China’s local economies [29,30]. On the other hand, with the growing trend of industrialization and anthropogenic activities, agricultural and forestland soils of southern China were contaminated with heavy metals [31]. This contamination increases the risk to bamboo plants and human health [31]. Therefore, many studies in this field are needed to find ways to reduce the toxicity of heavy metals and increase bamboo tolerance in these areas. This research aims to investigate the possible use of nitric oxide by the addition of sodium nitroprusside (SNP) as an NO synthesis agent to increase bamboo tolerance to heavy metal (such as Pb and Cd) toxicity by exploring the mechanisms involved in scavenging ROS compounds and stimulating plant antioxidant defense capacity.

## 2. Materials and Methods

### 2.1. Plant Preparation and In Vitro Condition

One-year-old bamboo plants (a single clone of *A*. *pygmaeus*) were chosen from the Bamboo Garden Nanjing Forestry University. For this experiment, long nodal explants (10 mm) of bamboo samples were provided. All of the nodal explants were cultured in vitro inside MS medium [32], which was allocated with 0.5 μM kinetin (KT), 4 μM 6-benzylaminopurine (6-BA), 8–10 g/L agar, and 25 g/L sucrose. After proliferation and shoot production, the shoots were transferred to glass petri dishes with a diameter of 60 mm by Murashige and Skoog (MS) medium for root production and proliferation. In this stage, MS medium was supplied with 4 μM nicotinic acid, 1.2 μM thiamine–HCl, 3 μM pyridoxine, 0.6 µM myo-inositol, 500 μM Pb and Cd alone or in combination with L-NAME, Hb, and SNP by 7–10 g/L agar and 30 g/L sucrose, which was adjusted to pH 5.8 ± 0.1. Additionally, 0.1 mg/L IAA (indole-3-acetic acid) was used as a regulator of growth hormone. In the next step, 1 L of MS medium was transferred to a microwave oven and sterilized at an optimum temperature of 120 °C for 30 min in an autoclave (HiClave HVE-50). The next day, the bamboo species in each treatment were cultured in a glass petri dish with a diameter of 60 mm and height of 90 mm, including 100 mL of culture medium in the inoculation hood (ultraviolet-sterilized -Air Tech) with a light of white fluorescent with special wavelengths with a range of (10–420 nm) at 20 °C for 4 h. The bamboo treatments were relocated to a plant tissue culture chamber with a photoperiod of 15 h and optimum temperatures between 18/23 °C in the dark and 28/24 °C in light periods for 30 days.

In this experiment, the treatments contained 200 µM Pb and 200 µM Cd in combination with 50 µM L-NAME, N(G)-nitro-L-arginine methyl ester (NO synthase inhibitor); 0.1% Hb, bovine hemoglobin (NO scavenger), including 200 µM SNP: NO donor in four replications in comparison to controls (Table 1).

### 2.2. Determination of Heavy Metal Accumulation and Nitric Oxide Concentrations in Shoots, Stems, and Roots

To determine the heavy metal (Pb and Cd) concentrations of shoots, stems, and roots, samples (0.5 g) were dissolved in 2 mL of H_2_O_2_ (30%) and 2 mL of HNO_3_ (67%). Then, the obtained solution was added to 10% KI, 10% HCl, and 5% ascorbic acid. To analyze Pb and Cd accumulation, the solution was transferred to an atomic absorption spectrometer (HG-AAS) (Shimadzu AA-6200) using optimal standards [33]. Nitric oxide concentrations were obtained by analyzing the conversion of oxygen-hemoglobin to methemoglobin. For this purpose, shoot, stem, and root samples (0.5 g) were dissolved in 3 mL of buffer (pH 6.0) with the addition of 0.1 M sodium acetate mixed with 1 M NaCl and 1% (*w/v*) ascorbic acid. Then, it was centrifuged for 30 min at 8000× *g* at 4 °C [34,35].

### 2.3. Measurement of Concentrations of Protein Thiol, Nonprotein, and a Total of Thiol

After homogenization of samples in an ice bath with 0.02 M EDTA, 0.5 mL of these samples was added to 1.8 mL of 0.2 M Tris buffer (pH 8.2) and 0.3 mL of 0.01 M 5,5′-dithiobis-(2-nitrobenzoic)(DTNB). Then, the content of the solution reached 10 mL with the addition of 7.9 mL of absolute methanol. In the last step, the amount of supernatant was recorded by measuring the absorbance at 412 nm after 10 min. To determine the total thiol (TTs), this experiment was conducted with 13.1 mM^−1^ cm^−1^ as the extinction coefficient. To determine the nonprotein thiols (NPTs), homogenates (5 mL) were added to 1 mL of 50% TCA and 4 mL distilled water for 10 min. Then, 2 mL of the obtained supernatant was mixed in 0.2 mL of DTNB and 5 mL of 0.4 M Tris buffer (pH 8.9). In the final step, for determining the nonprotein thiols (NPTs), the obtained absorbance was recorded in 5 min at 412 nm in a spectrometer machine. Protein thiols (PTs) were obtained by subtracting total thiols (TTs) from NPTs [36].

### 2.4. Quantification of Glycine Betaine (GB), Proline Contents (Pro), and GSH

The content of glycine betaine was obtained using the Grieve and Grattan [37] method, which can calculate the obtained absorbance at 365 nm by using a standard curve of GB. The proline content was measured by Bates et al. (1973) [38]; for this purpose, 330 mg samples (leaves) were digested by the optimum amount of sulfosalicylic acid. Then, the obtained supernatant was transferred to a spectrometer machine. The absorbance was measured at 520 nm, and the proline content was determined using a standard curve. The concentrations of GSH were estimated based on the methods used by Ellman (1959) [39]. Therefore, 0.1 g of bamboo sample was digested in 4 mL of 0.5 mM EDTA solution comprising 3% TCA at 4 °C. The mixture was centrifuged at 13,000× *g* for 20 min. Then, 0.2 mL of supernatant was mixed with 0.2 mM 5,5-dithiol-bis (2-nitrobenzoic) (DTNB) and 1.5 mL of 50 mM potassium phosphate buffer (pH = 6.9). The solution was held at a temperature of 30 °C for 2 min. Then, glutathione (GSH) was recorded at an absorbance of 412 nm.

### 2.5. Determination of Lipid Peroxidation (MDA), Superoxide Radical (O2^•−^), Hydrogen Peroxide (H_2_O_2_), Soluble Proteins (SP), and Electrolyte Leakage (EL)

Lipid peroxidation as malondialdehyde (MDA) content was measured to demonstrate cell lipid peroxidation, which was conducted according to the methods used by Madhava Rao and Sresty [40]. For this purpose, the optical density (OD) was estimated by absorbance at 532 and 600 nm. Hydrogen peroxide (H_2_O_2_) was conducted by the method of Velikova et al. (2000) [41], such that the samples (leaf) were digested in trichloroacetic acid (0.1%), and then, the obtained supernatant was added to 1 M potassium iodide and potassium phosphate buffer (100 mM, pH 7.0). OD was measured at an absorbance of 390 nm. The superoxide radical (O2^•−^) was assessed by Li’s method [42]. In this experiment, leaf tissue samples (200 mg) were digested in phosphate buffer (65 mM, pH = 7.8) and then centrifuged at 4000× *g* for 20 min. The supernatant was incubated in phosphate buffer (65 mM, pH = 7.8) and 10 mM hydroxylamine hydrochloride for 15 min at the optimum temperature of 27 °C. Then, the solution was added to 17 mM sulfanilamide and 7 mM α-naphthylamine, which was held for 20 min and then measured at 530 nm at 25 °C. The final step for measuring the rate of O2^•−^ used a standard curve with a nitrogen dioxide radical (NO_2_). The soluble protein was measured by the Bradford method [43]. Hence, the soluble protein content was estimated by the effect of Coomassie Brilliant Blue (G25) on changes in protein levels, which was recorded by using a spectrometer. Electrolyte leakage (EL) was determined according to the method of Valentovic et al. (2006) [44]. At the core of this method, 15 mL of deionized water was added to 0.3 g of leaf samples. Then, the solution was held at the optimum temperature of 25 °C for 2 h. Then, the primary electrical conductivity (EC1) of the solution was measured. In the next step, the samples were preserved in one autoclave at 120 °C for 17 min. Then, the secondary electrical conductivity (EC2) was measured again. The EC was determined as follows:EL (%) = (EC1/EC2) × 100.

### 2.6. Determination of Antioxidant Activities

The leaf samples (0.5 g) were washed carefully, cut with scissors, and ground in liquid nitrogen. Then, the obtained powder was added to phosphate-buffered saline (pH 7.2–7.4) at the optimum temperature of 2–8 °C and centrifuged at 2500–3000× *g* for 15 min. The obtained supernatant was used for the measurement of antioxidant enzyme activities.

The activity of superoxide dismutase (SOD) was estimated by the Dhindsa and Matowe (1981) [45] method, which is based on the photochemical reduction of nitroblue tetrazolium (NBT) and was recorded at 560 nm. Peroxidase (POD, E.C. 1.11.1.7) was determined according to the Zhang method [46]. Therefore, in these indices, the OD amount was measured according to the alternation in absorbance at 470 nm. Catalase (CAT) activity was recorded at 240 nm by the Aebi method [47]. The activity of glutathione reductase (GR) as EU mg^−1^ protein was determined based on Foster and Hess’s method [48], which was recorded at 340 nm. The activity of ascorbate peroxidase (APX) as EU mg^−1^ protein was measured at 290 nm by the Nakano and Asada method [49]. The activity of glutathione S-transferase (GST) (EC 2.5.1.18) was determined spectrophotometrically (Rohman et al., 2009) [50].

### 2.7. Determination of Photosynthetic Pigments (Chlorophyll and Carotenoids)

Photosynthetic pigments, including chlorophyll-a, chlorophyll-b, total chlorophyll, and carotenoids, were determined by the Lichtenthaler method [51]. Based on this method, bamboo leaves (0.5 g) were digested in a mortar with the addition of liquid nitrogen. The obtained powder was mixed with 20 mL of 80% acetone at 0 to 5 °C. Then, the samples were centrifuged at 6000× *g* for 10 min. The obtained supernatant was measured at absorbances of 470, 645, and 663 nm for carotenoid, chlorophyll b, and chlorophyll a content, respectively. The obtained data were determined by using the following formulas and represented in units of mg/g fresh weight:Chlorophyll a = 12.25A663 − 2.79A647
Chlorophyll b = 21.50A647 − 5.10A663
Total Chlorophyll = Chlorophyll a + Chlorophyll b
Carotenoid = 1000A470 − 1.82Chl a − 95.15Chl b/225

The assay of the translocation factor (TF), tolerance index (TI), and bioaccumulation factor (BF) was assessed.

In this study, the indices of TF, TI, and BF in the shoots were calculated according to the method of Souri and Karimi [52], which used photoextraction for its efficiency. The values of TF and TI (shoot and root) and BF (shoot) were obtained by the following formula.
$$\mathrm{TF}=\frac{\text{Concentrations of heavy metals in the shoot}}{\text{Concentrations of heavy metals in the root}}$$
$$\mathrm{TI}(\mathrm{shoot})=\frac{\text{shoot dry weight of heavy metals treatments}}{\text{shoot dry weight of control treatment}}$$
$$\mathrm{TI}(\mathrm{root})=\frac{\text{root dry weight of heavy metals treatments}}{\text{root dry weight of control treatment}}$$
$$\mathrm{BF}(\mathrm{shoot})=\frac{\text{Concentrations of heavy metals in the shoot}}{\text{Concentrations of heavy metals in the medium}}$$

### 2.8. Determination of Shoot Dry Weight, Root Dry Weight, and Shoot Length

After washing the plant roots and shoots, they were transferred to an oven (vacuum drying oven (DZF-6090)) at 100 °C for 25 min. Then, the samples were dried at an optimum temperature of 70 °C under constant dry weight condition. After the end of the drying process, they were used for the determination of biomass (shoot DW and root DW). Each treatment was replicated four times. For the determination of shoot length, they were measured two times, one at the beginning and one at the end of the study.

### 2.9. Statistical Analysis

The data analysis was conducted under a completely randomized design (CRD) through a 2-way factorial design with four replicates. The statistical package of R software was allocated to the analysis of variance (ANOVA). Tukey’s test was supplied for the comparison of the mean difference at the *p* < 0.05 probability level.

## 3. Results

### 3.1. NO Donors (SNPs) Reduce Pb and Cd Accumulation in Shoot, Stems, and Roots in Plants under Heavy Metals

The accumulation of heavy metals in plant organs (shoot, stems, and roots) is one of the important factors in increasing plant toxicity and can be a major obstacle to plant growth and yield. In the present study, the results showed that SNPs at high concentrations in the roots, shoots, and stems can reduce the levels of heavy metals in plant organ surfaces significantly. This reduction indicates that SNP has the ability to detoxify plants. According to Table 2, SNP in combination with 200 μM Pb and 200 μM Cd reduces the concentration of heavy metals by 64% and 57% in the shoots, 66% and 52% in the stems, and 49% and 42% in the roots in comparison with the control treatments. On the other hand, the treatment of 50 µM L-NAME + 1% Hb increases the levels of heavy metal with a 1.61-fold and 1.47-fold reduction in the shoot, 1.66 and 1.57-fold reduction in the stem, and 1.51-fold and 1.47-fold reduction in the root comparison with control, respectively. We suggest that the addition of SNPs as NO donors in plants can reduce heavy metal (Pb and Cd) concentrations in combination with L-NAME + 1% Hb.

### 3.2. The NO Donor (SNP) Increased the Protein, Non-Protein, and Total Thiol Contents in Plants under Heavy Metal Toxicity (Pb and Cd)

With the investigation of the influencing principal compounds of NO on proteins, non-proteins, and total thiols, we observed that there was a significant difference between the various Pb and Cd treatments (*p* < 0/001) (Figure 1), which showed that 200 µM SNP has the ability to increase thiol in combination with L-NAME and Hb. This study demonstrated that most of the enhancement was related to the 200 µM SNP treatment, 200 µM SNP+ 200 µM Pb treatment, and 200 µM SNP + 200 µM Cd treatment with 39%, 41%, and 41% increases in protein content; 30%, 37%, and 38% increases in non-protein content; and 50%, 49%, and 49% increases in total thiol compared with the control treatment, respectively (Figure 1). On the other hand, the results showed that the combination of Pb or Cd with L-NAME and Hb significantly reduced the content of protein, non-protein, and total thiol in bamboo plants in comparison with the control, which demonstrates the positive role of NO in improving protein and non-protein contents under stress conditions.

### 3.3. NO Donors (SNPs) Improve Membrane Injury and Increase the Cell Protection of Bamboo Plants under Heavy Metal Toxicity (Pb and Cd)

In the present study, to calculate the impact of NO on ROS compounds, the cell membrane, and plant lipoperoxidation, hydrogen peroxide (H_2_O_2_), superoxide radicals (O2^•−^), soluble protein (SP), lipid peroxidation (MDA), and electrolytes leakage (EL) were measured. The results show that there is a significant difference between the various Pb and Cd treatments in combination with the NO compound (*p* < 0.001). Based on the obtained result, while the combination of L-NAME and Hb increases hydrogen peroxide (H_2_O_2_), superoxide radical (O2^•−^), soluble protein (SP), electrolyte leakage (EL), and lipid peroxidation (MDA), the addition of an NO donor (SNP) reduces the content of hydrogen peroxide (H_2_O_2_), superoxide radical (O2^•−^), soluble protein (SP), lipid peroxidation (MDA), and electrolyte leakage (EL) compared with the control treatment. This reduction shows the key role of NO in improving the cell membrane and protecting plant cells from the severe impact of ROS compounds in plants under 200 µM Pb and 200 µM Cd. Hence, the greatest reduction of ROS compound and cell membrane indicators are related to a combination of 200 µM SNP alone and in combination with 200 µM Pb and 200 µM Cd with 33%, 29%, and 30% reduction in hydrogen peroxide (H_2_O_2_); 59%, 46%, and 45% reduction in superoxide radical (O2^•−^); 44%, 35%, and 30% reduction in soluble proteins (SP); 55%, 53%, and 55% reduction in lipid peroxidation (MDA); and 59%, 48%, and 47% reduction in electrolyte leakage (EL) in comparison with the control treatment. We suggest that NO donors (SNPs) with increasing antioxidant activity reduce the generation of ROS compounds in the bamboo species under heavy metals (Pb and Cd) (Figure 2).

### 3.4. The NO Donor (SNP) Increased Glycine Betaine and Proline Contents and GSH Content in Plants Experiencing Heavy Metal Toxicity (Pb and Cd)

The results show that the NO donor (SNP) significantly increases the contents of proline and glycine betaine and the GSH content in plants under Pb and Cd (*p* < 0.001). In this study, the results demonstrate that L-NAME and Hb in combination with heavy metals (Pb and Cd) reduces proline, glycine betaine, and GSH contents. Therefore, the lowest amount of proline, glycine betaine, and GSH is related to the combination of 200 µM Pb + 50 µM L-NAME + 0.1% Hb and 200 µM Cd + 50 µM L-NAME + 0.1% Hb with 52% and 53% reductions in proline content; 64% and 55% reductions in glycine betaine; and 47% and 51% reductions in GSH content in comparison with the control, respectively. On the other hand, the results of data analysis show that the highest proline content, glycine betaine content, and GSH content are related to the levels of NO donors (SNPs) alone or in combination with 200 µM Pb and 200 µM Cd, with 1.31-fold, 1.29-fold, and 1.30-fold increases in proline content; 1.39-fold, 1.44-fold, and 1.47-fold increases in glycine betaine; and 1.55-fold, 1.61-fold, and 1.68-fold increases in GSH content, respectively. This association shows the key role of NO donors (SNPs) in glycine betaine, proline contents, and GSH content in plants under heavy metal toxicity (Pb and Cd) (Table 3).

### 3.5. NO Donors (SNPs) Improve Antioxidant Enzyme Activities in Bamboo Plants under Heavy Metals (Pb and Cd)

The response of L-NAME, Hb, and SNP to plant antioxidant activity under 200 µM Pb and 200 µM Cd is shown in Figure 3. According to the results, there was one significant difference between the separate combinations of NO and various levels of Pb and Cd (*p* < 0.001). The results show that antioxidant activity decreases with the addition of L-NAME and Hb to heavy metals (Pb and Cd) in comparison with the control, while the addition of SNP to 200 µM Pb and 200 µM Cd increases antioxidant activity in comparison to the control. Additionally, the results show that SNP improves antioxidant capacity in combination with heavy metals (Pb, Cd)–L-NAME and heavy metals (Pb, Cd) -Hb in comparison with the control treatment. However, the combination of heavy metals (Pb and Cd)-L-NAME-Hb and SNP together does not increase antioxidant activity in bamboo species in comparison with the control. These results indicate that SNPs have the ability to stimulate antioxidant activity in plants under heavy metal stress (Pb and Cd). This ability revealed that the greatest enhancement in antioxidant enzyme activity is related to the combination of SNP alone and in combination with two heavy metals (Pb and Cd) by a 1.18-fold, 1.14-fold, and 1.14-fold enhancement in SOD activity; 1.34-fold, 1.31-fold, and 1.35-fold enhancement in POD activity; 1.27-fold, 1.26-fold, and 1.30-fold enhancement in CAT activity; 1.38-fold, 1.45-fold, and 1.44-fold enhancement in GR activity; 1.36-fold, 1.24-fold, and 1.25-fold enhancement in APX activity; and 1.55-fold, 1.55-fold, and 1.54-fold enhancement in GST activity in comparison with control treatments, respectively. In contrast, the results indicate that a combination of Pb-L-NAME-Hb and Cd-L-NAME-Hb significantly reduce antioxidant activity with a 24% and 21% reduction in SOD activity, 66% and 66% reduction in POD activity, 44% and 48% reduction in CAT activity, 72% and 70% reduction in GR activity, 59% and 62% reduction in APX activity, and 58% and 53% reduction in GST activity in comparison with the control treatment. We suggest that SNPs as NO donors improve the plant defense system under heavy metal (Pb and Cd) toxicity (Figure 3).

### 3.6. NO Donors (SNPs) Improve Photosynthetic Pigments (Chlorophyll and Carotenoids) in Plants under Heavy Metal Toxicity (Pb and Cd)

The contents of chlorophyll and carotenoids can be a strong indicator of photosynthesis efficiency in plants. Hence, photosynthetic pigments (chlorophyll and carotenoids) were measured in bamboo plants under heavy metal stress. The results obtained by data analyses indicate that there is one significant difference between the various concentrations of NO with 200 μM Pb and 200 μM Cd (*p* < 0.001). According to the results, the addition of an NO donor (SNP) improves the chlorophyll and carotenoid contents in plants under the combination of heavy metals + L-NAME + 1% Hb. Therefore, the SNP alone and in combination with heavy metals (Pb and Cd) shows the highest chlorophyll and carotenoid contents, with 18%, 15%, and 18% increases in chlorophyll-a; 39%, 44% and 43% increases in chlorophyll-b; 31%, 26% and 42% increases in total chlorophyll; and 46%, 72% and 43% increases in carotenoid contents, respectively, in comparison to the control treatments. On the other hand, the combination of heavy metals (Pb and Cd) + L-NAME +1% Hb shows the lowest amount of chlorophyll-a, chlorophyll-b, total chlorophyll and carotenoids, with 33% and 39% reductions in chlorophyll-a, 42% and 50% reductions in chlorophyll-b, 38% and 34% reductions in total chlorophyll, and 22% and 41% reductions in carotenoid contents in comparison to the control treatments. We suggest that NO has a strong ability to increase chlorophyll content and carotenoids in bamboo plants experiencing heavy metal (Pb and Cd) toxicity (Table 4).

### 3.7. NO Donors (SNPs) Improve Plant Growth and Biomass under Pb and Cd Toxicity

Plant biomass (the dry weight of roots and shoots) and shoot length were measured as indicators of plant growth under heavy metal toxicity. According to the results, there was a significant difference between various concentrations of nitric oxide alone and in combination with two heavy metals (Pb and Cd) (*p* < 0.001). Therefore, the results suggest that SNP can increase shoot length and plant biomass under Pb and Cd toxicity. On the other hand, the results show that the concentration of L-NAME + 1% Hb as an NO synthase inhibitor and NO scavenger significantly reduces shoot length and plant biomass. This shows the role of NO in plant growth under heavy metal toxicity. In this study, the greatest increase in DW root, DW shoot, and shoot length is related to the individual levels of SNP and in combination with heavy metals (Pb and Cd), with a 31%, 31%, and 33% increase in DW shoot; 26%, 25%, and 26% increase in DW root; and 17%, 12%, 12% increase in the shoot length in comparison with control treatment, while the lowest amount of them was observed in the combination of 50 µM L-NAME + 1% Hb + 200 µM Pb and 50 µM L-NAME +1 % Hb + 200 µM Cd with a 45% and 47% reduction in DW shoot, 46% and 52% reduction in DW root, and 30% and 32% reduction in shoot length, respectively (Figure 4).

### 3.8. NO Donors (SNPs) Reduce the Translocation Factor (TF) and Bioaccumulation Factor in Shoots, as Well as Increase the Tolerance Index (TI) in the Shoots and Roots of Bamboo Species under 200 μM Pb and 200 μM Cd

To determine the involved mechanisms of bamboo plant exposure to heavy metal toxicity (Pb and Cd), the translocation factor (TF) was calculated. The obtained data show that SNPs play a key role in the reduction of heavy metal translocation from roots to shoots, as shown in Table 5. According to data, SNPs indicate 56%, 51%, and 51% reductions in the translocation of heavy metals (Pb and Cd) from shoots to roots, respectively. On the other hand, the bioaccumulation factor (BF) in plant shoots was calculated. amd according to the results, SNP significantly reduces BF in shoots by a 70% and 63% reduction under 200 μM Pb and 200 μM Cd in comparison with the control treatment. This paper aimed to assess the impact of nitric oxide on bamboo plant tolerance under heavy metal (Pb and Cd) stress. Hence, the tolerance index (TI) was calculated. According to the data in Table 5, the addition of sodium nitroprusside (SNP) as an NO donor can increase the tolerance index (TI) in plant organs under 200 μM Pb and 200 μM Cd by 25% and 21% increases in shoots and 19% and 16% increases in roots, respectively, compared with their control treatments. Therefore, according to Table 2 and Table 5, we suggest that NO, with a reduction in heavy metal (Pb and Cd) accumulation and heavy metal (Pb and Cd) translocation, increases plant tolerance for heavy metal toxicity. This reduction ability can introduce bamboo species as a good option for phytoremediation technology and environmental safety.

## 4. Discussion

Nitric oxide (NO) is an endogenous signaling molecule that can act on gene transcription and adjust many processes through cell membranes [53]. Many studies have reported that NO, as an exogenous application, increases plant tolerance to various abiotic stresses. Previous scholars expressed that this tolerance is related to the key role of NO in scavenging ROS by increasing antioxidant activity in various plants [19,52,54,55,56,57]. In fact, NO protects plants under heavy metal toxicity by signaling, plant perception, and stress acclimation [56]. In this study, SNPs, as NO donors, increased antioxidant capacity (SOD, CAT, POD, GR, APX, and GST) in plants under heavy metal (Pb and Cd) stress, which led to scavenging ROS compounds (H_2_O_2_ and O_2_), improving membrane injury with a reduction in soluble proteins (SP) and electrolytes leakage (EL), and finally, ameliorating cell oxidation in bamboo plants [57]. Therefore, we suggest that exogenous application of NO acts as a signaling molecule to regulate cell membranes and protect cells in plants experiencing heavy metal (Pb and Cd) stress. The effective role of nitric oxide in the amelioration of membrane peroxidation and reduction of ion leakage has been demonstrated in previous studies [58,59]. On the other hand, our results indicated that two factors, bovine hemoglobin (Hb) as an NO scavenger and N(G)-nitro-L-arginine methyl ester (L-NAME) as an NO synthase inhibitor, increase oxidative stress in bamboo plants with the generation of ROS compounds and reduce antioxidant activities. However, the addition of SNP as an NO donor increases antioxidant activity and reduces cell oxidation in the plant. Therefore, we suggest that NO, with the regulation of cells and phytoglobins1 (nonsymbiotic hemoglobins1, nsHbs1), ameliorates the negative effect of Hb on the generation of ROS compounds [60]. The role of nitric oxide in the scavenging of ROS compounds has been proven by many studies [56,61,62,63].

It has been reported that NO can prevent carbonylation of thiol proteins by ROS [64]. Our results demonstrate that SNPs have the ability to increase protein thiol, non-protein thiol, and total thiol in plants under heavy metal (Pb and Cd) toxicity. On the other hand, the results indicate that the application of Hb and L-NAME in combination with heavy metals (Pb and Cd) reduces TT, PT, and NPT, which have demonstrated the role of NO in the regulation of thiol proteins and non-proteins in plants under stress. Proline, an osmoprotectant, prevents macromolecule dehydration, and proline, an antioxidant factor, scavenges ROS, which can increase plant resistance to environmental stress [65]. On the other hand, proline can help to improve antioxidant activity in plants under stress [66], which was revealed in our study. Therefore, according to our results, SNPs increase proline content under heavy metal (Pb and Cd) stress, which occurs by regulating proline synthesis by regulating P5CS1, which encodes 1-pyrroline 5-carboxylate synthetase [67,68]. GSH is a low molecule that utilizes a system of buffering against redox imbalance, which can be found in all plant organelles. GSH plays an important role in the amelioration and reduction of plant stress [69]. The results show that SNP increases GSH contents in combination with heavy metals (Pb and Cd) and applications of L-NAME and Hb. The positive impact of proline and GSH on the amelioration of heavy metal stress has been proven by many studies [70,71]. GB content is another important compatible substance that plays a vital role in plant osmotic adjustment under stress conditions [19], and there is some evidence that shows that the accumulation of GB increases photosynthetic pigments. Therefore, most GBs accumulate in chloroplasts and indirectly improve safeguarding performance in photosystem II (PSII). Hence, GB can help increase plant photosynthesis efficiency under stressful conditions. The increasing chlorophyll content in plants can be an indicator of photosynthesis performance in plants [72]. Our results demonstrate that N(G)-nitro-L-arginine methyl ester (L-NAME), an NO synthase inhibitor, and bovine hemoglobin (Hb), an NO scavenger, reduce chlorophyll and carotenoid contents in plants. The addition of SNP improves chlorophyll and carotenoid contents in the plants under heavy metal (Pb and Cd) stress. This result shows the positive role of NO in increasing photosynthesis indices. We suggest that NO increases antioxidant activity and that GB accumulation could improve photosynthetic properties in plants under toxicity. NO is a key signaling molecule that leads to improved plant growth under several abiotic stresses [73], which has been demonstrated in our study. Our results demonstrate the addition of SNP as one NO donor that can improve plant growth indices, including DW shoot, DW root, and shoot length in plants under heavy metal (Pb and Cd) toxicity, which is in line with other studies [74,75,76]. The positive impact of NO on plant growth can be ascribed to increasing antioxidant activity, reducing plant osmolytes, membrane ion leakage, and heavy metal accumulation in plant organs [19]. This positive impact can, finally, lead to an increase in plant photosynthesis and plant growth under heavy metal toxicity. The mechanisms involved in the analyses of NO functions are still not completely clear, but there is some evidence suggesting that NO can increase plant resistance to heavy metal toxicity by reducing heavy metal accumulation and ameliorating oxidative stress in plants [61,62,63,64,65,66,67,68,69,70,71,72,73,74,75,76,77]. Some parameters, such as bioaccumulation factor (BF), the tolerance index (TI), and translocation factor (TF) evaluate plant phytoremediation potential [78,79,80]. Our results significantly indicate that the application of SNP reduces Pb and Cd accumulation in the shoots, stems, and roots, which leads to reduced metal accumulation in this study. The reduction of heavy metals by NO has been reported by many studies and in many plant species, such as barley [81], rice [65] and *Isatis cappadocica* [17]. On the other hand, the reduction in heavy metal accumulation in shoots and stems is related to the reduction in uptake and/or heavy metal translocation from roots to shoots by SNP [78]. According to Table 5, SNPs in combination with heavy metal (Pb and Cd) stress increase the tolerance index (TI) in shoots and roots. It has been reported that TF values less than one indicate an insufficient metal transfer system in a plant, which means that the plant is willing to accumulate heavy metals in rhizomes and roots more than aerial parts, such as shoots and leaves [82]. The results show that SNP in combination with heavy metals (Pb and Cd) reduce TF to less than one, which shows a 56% reduction in comparison with the control treatments. This combination also increases the tolerance index in our bamboo species. According to the results, we concluded that NO plays an important role in increasing bamboo tolerance for excess heavy metal toxicity. This increased tolerance can help bamboo plant use as phytoremediation plants to ameliorate environmental pollution. There are many plants that act as hyperaccumulators and can be considered to be used in phytoremediation technology. However, not every hyperaccumulator is a suitable candidate for phytoremediation purposes. In fact, the reports show that very few plants meet all the hyperaccumulator features that are required for phytoremediation. For instance, there are hyperaccumulators with limited root extraction capacities, but they may compensate for this shortcoming by having rapid vegetative growth or producing high biomass [83]. Yet, such hyperaccumulators are not perfect for phytoremediation. Recent studies have reported that some bamboo species have a high ability to adapt to metalliferous environments and a high capacity to absorb heavy metals [84,85,86]. As a matter of fact, many bamboo plants due to their morphological growth characteristics, such as extensive root system and rapid vegetative multiplication, as well as high biomass content, are suitable for phytoremediation purposes. In particular, the roots, which are the main site of absorption of excess heavy metal by bamboo plants, are capable of transferring the absorbed metals to the aerial parts only when they have already taken up large quantities of heavy metals. Bamboo tissues in the rhizome and culm can accumulate a large number of heavy metals that mainly accumulate in the cell wall, vacuole, and cytoplasm [83]. According to the results shown in Table 2, the heavy metals accumulation value of bamboo in the roots is higher than the stem and the shoots, indicating that bamboo has a strong ability in accumulation of heavy metals in the root surface. This shows the important role that bamboo plants can play in phytoremediation, which is confirmed by another study [87]. On the other hand, our findings show that the application of NO can increase the bamboo tolerance by a significant reduction of bioaccumulation factor (BF) in the shoot, as well as by a diminished heavy metal translocation from root to shoot, with an eventual increase in tolerance factor (TF) in the shoot and the root of bamboo species (Table 5). Therefore, our results unequivocally demonstrated that NO helps plant phytoremediation in the accumulation of heavy metals on the root surface.

## 5. Conclusions

The results obtained by this study revealed the protective role of nitric oxide as a signaling molecule in the amelioration of the severe impact of heavy metal toxicity in plants. We found that sodium nitroprusside, an NO donor, increased the phytoremediation efficiency of bamboo species under heavy metal (Pb and Cd) toxicity, which was achieved by some key mechanisms, such as increasing antioxidant capacity, reducing metal accumulation, inhibiting metal translocation from roots to shoots, and finally, improving plant photosynthesis and plant growth. Additionally, the results showed that SNPs protect the plant cell membrane by scavenging ROS compounds, lipid peroxidation, and regulating ion leakage. In contrast, N(G)-nitro-L-arginine methyl ester (L-NAME), a nitric oxide synthase inhibitor and 0.1% bovine hemoglobin (Hb), a nitric oxide scavenger, demonstrated a negative impact on plant growth promotion of antioxidant enzyme activity, protein, and non-protein; total thiol, glycine betaine, and proline contents; and GSH content, heavy metal accumulation and translocation, and plant tolerance under metal stress. We suggest that the exogenous application of NO can increase plant tolerance for heavy metal toxicity. However, different plant species need to be used in future studies. We concluded that nitric oxide (NO) can play a role in plant perception, signaling, and heavy metals acclimation. Moreover, NO increased the accumulation of the heavy metals in the root surface, leading to increased phytoextraction of the heavy metals by the plants from the soil. As a consequence, it can help to improve the phytoremediation potential of bamboo plants in the environment.

## Figures and Tables

**Figure 1 antioxidants-10-01981-f001:**
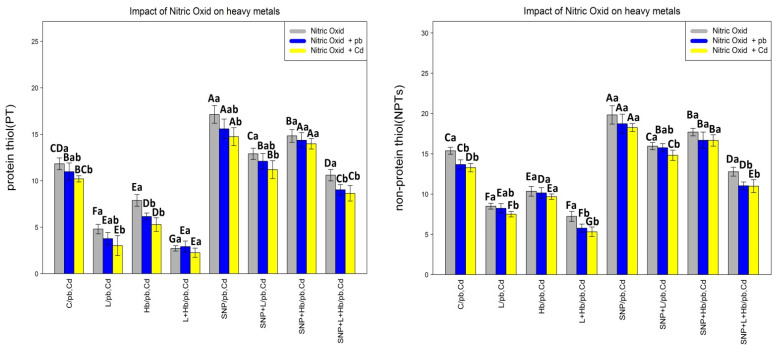
Effect of nitric oxide (NO) concentrations on protein, non-protein, and a total of thiol of *A. pygmaea* L. with 200 μM Pb and 200 μM Cd. The treatments contained various concentrations of nitric oxide alone or in combination with 200 μM Pb and 200 μM Cd. The capital letters showed statistically significant differences across various levels of nitric oxide alone or in combination with 200 μM Pb and 200 μM Cd (the bars with the same colors), while the lowercase letters showed statistically significant differences within each concentration of nitric oxide (NO) alone or in combination with 200 μM Pb and 200 μM Cd (the bars with different colors) according to Tukey’s test (*p* < 0.05).

**Figure 2 antioxidants-10-01981-f002:**
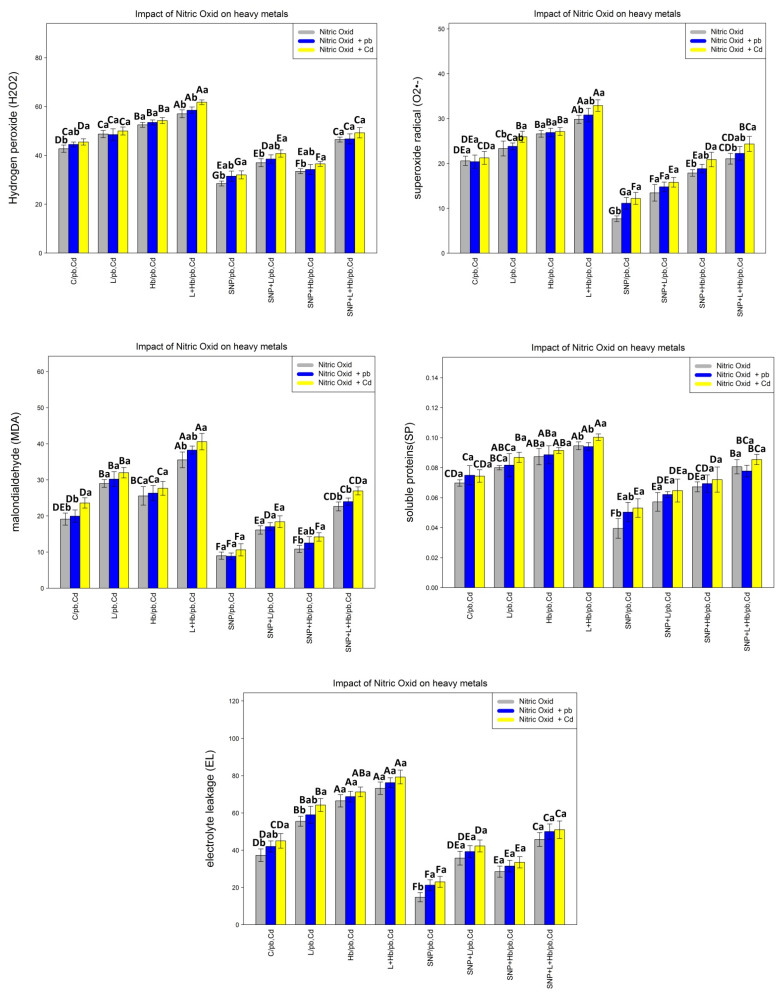
Effect of nitric oxide (NO) concentrations on hydrogen peroxide (H_2_O_2_), superoxide radical (O2^•−^), soluble proteins (SP), lipid peroxidation (MDA), and electrolyte leakage (EL) of *A. pygmaea* L. with 200 μM Pb and 200 μM Cd. The treatments contained various concentrations of nitric oxide alone or in combination with 200 μM Pb and 200 μM Cd. The capital letters showed statistically significant differences across various levels of nitric oxide alone or in combination with 200 μM Pb and 200 μM Cd (the bars with the same colors), while the lowercase letters showed statistically significant differences within each concentration of nitric oxide (NO) alone or in combination with 200 μM Pb and 200 μM Cd (the bars with different colors) according to Tukey’s test (*p* < 0.05).

**Figure 3 antioxidants-10-01981-f003:**
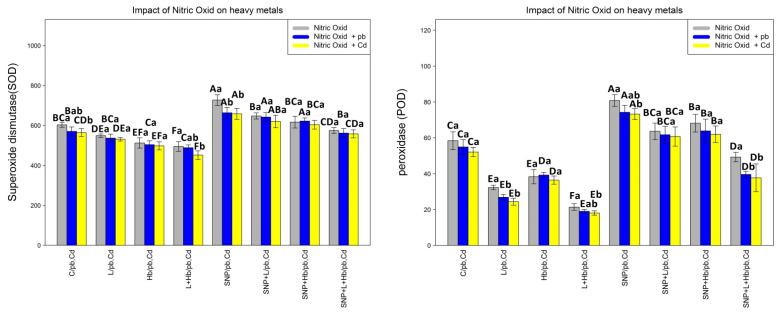
Effect of nitric oxide (NO) concentrations on antioxidant enzyme activities (SOD, POD, CAT, GR, APX and GST), of *A. pygmaea* L. with 200 μM Pb and 200 μM Cd. The treatments contained various concentrations of nitric oxide alone or in combination with 200 μM Pb and 200 μM Cd. The capital letters show statistically significant differences across various levels of nitric oxide alone or in combination with 200 μM Pb and 200 μM Cd (the bars with the same colors), while the lowercase letters show statistically significant differences within each concentration of nitric oxide (NO) alone or in combination with 200 μM Pb and 200 μM Cd (the bars with different colors) according to Tukey’s test (*p* < 0.05).

**Figure 4 antioxidants-10-01981-f004:**
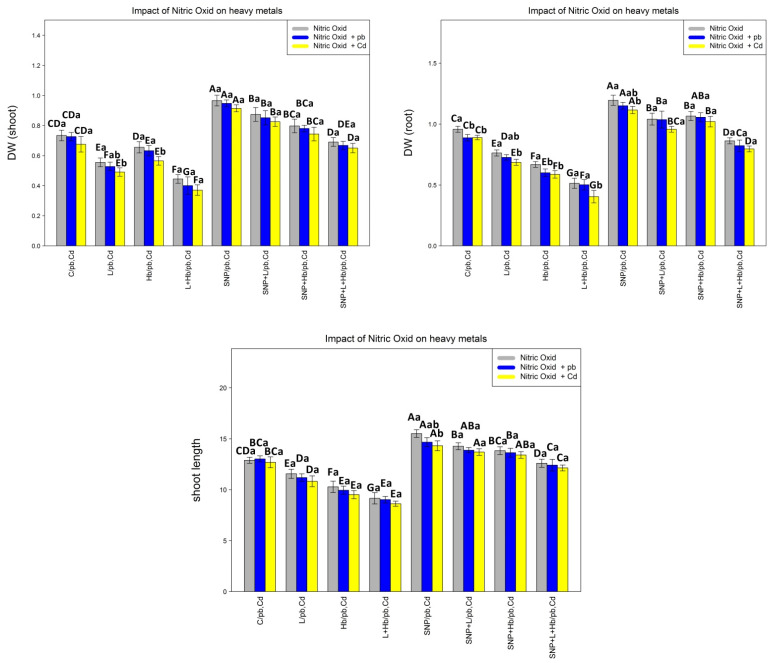
Effect of nitric oxide (NO) concentrations on bamboo biomass (DW shoot and DW root) and shoot length of *A. pygmaea* L. with 200 μM Pb and 200 μM Cd. The treatments contained various concentrations of nitric oxide alone or in combination with 200 μM Pb and 200 μM Cd. The capital letters showed statistically significant differences across various levels of nitric oxide alone or in combination with 200 μM Pb and 200 μM Cd (the bars with the same colors), while the lowercase letters showed statistically significant differences within each concentration of nitric oxide (NO) alone or in combination with 200 μM Pb and 200 μM Cd (the bars with different colors) according to Tukey’s test (*p* < 0.05).

**Table 1 antioxidants-10-01981-t001:** The experimental design.

Treatments	Concentrations
Control	0
Pb	200 µM Pb
Cd	200 µM Cd
L-NAME	50 µM L-NAME
L-NAME + Pb	50 µM L-NAME + 200 µM Pb
L-NAME + Cd	50 µM L-NAME + 200 µM Cd
Hb	0.1% Hb
Hb + Pb	0.1% Hb + 200 µM Pb
Hb + Cd	0.1% Hb + 200 µM Cd
L-NAME + Hb	50 µM L-NAME + 0.1% Hb
L-NAME + Hb + Pb	50 µM L-NAME + 0.1% Hb + 200 µM Pb
L-NAME + Hb + Cd	50 µM L-NAME + 0.1% Hb + 200 µM Cd
SNP	200 µM SNP
SNP + Pb	200 µM SNP + 200 µM Pb
SNP + Cd	200 µM SNP + 200 µM Cd
SNP + L-NAME	200 µM SNP + 50 µM L-NAME
SNP + L-NAME + Pb	200 µM SNP + 50 µM L-NAME + 200 µM Pb
SNP + L-NAME + Cd	200 µM SNP + 50 µM L-NAME + 200 µM Cd
SNP + Hb	200 µM SNP + 0.1% Hb
SNP + Hb + Pb	200 µM SNP + 0.1% Hb + 200 µM Pb
SNP + Hb +Cd	200 µM SNP + 0.1% Hb + 200 µM Cd
SNP + L-NAME + Hb	200 µM SNP + 50 µM L-NAME + 0.1% Hb
SNP + L-NAME + Hb + Pb	200 µM SNP + 50 µM L-NAME + 0.1% Hb + 200 µM Pb
SNP + L-NAME + Hb + Cd	200 µM SNP + 50 µM L-NAME + 0.1% Hb + 200 µM Cd

Pb: (lead), Cd: (cadmium), L-NAME: N(G)-nitro-L-arginine methyl ester (NO synthase inhibitor), Hb: bovine hemoglobin (NO scavenger), and SNP: NO donor.

**Table 2 antioxidants-10-01981-t002:** The accumulation levels of different concentrations of nitric oxide and heavy metal (Pb, Cd) contents in bamboo shoots, stems, and roots.

Nitric Oxid Concentration	Heavy Metals Content	Heavy Metal Accumulation (Shoot)	Nitric Oxide Accumulation (Shoot)	Heavy Metal Accumulation (Stem)	Nitric Oxide Accumulation (Stem)	Heavy Metal Accumulation (Root)	Nitric Oxide Accumulation (Root)
µmol/L	µmol/L	µg/L	µg/L	µg/L	µg/L	µg/L	µg/L
0	0	0	0	0	0	0	0
0	200 µM Pb	21.35 ± 0.61 ^Eb^	0	23.00 ± 0.81 ^Eb^	0	32.87 ± 0.47 ^Db^	0
0	200 µM Cd	24.35 ± 0.38 ^Da^	0	25.25 ± 0.50 ^Ea^	0	35.90 ± 0.40 ^Da^	0
50 µM L-NAME	0	0	15.55 ± 0.45 ^Da^	0	19.65 ± 0.90 ^Ea^	0	25.60 ± 0.35 ^Ea^
50 µM L-NAME	200 µM Pb	26.17 ± 1.11 ^Cb^	14.15 ± 0.50 ^Eb^	28.26 ± 0.93 ^Cb^	16.87 ± 0.53 ^Eb^	39.61 ± 1.00 ^Cb^	22.40 ± 0.82 ^Eb^
50 µM L-NAME	200 µM Cd	27.75 ± 0.52 ^Ca^	12.75 ± 0.50 ^Ec^	30.12 ± 0.83 ^Ca^	16.37 ± 0.81 ^Eb^	42.55 ± 0.43 ^Ca^	19.92 ± 1.09 ^Ec^
0.1% Hb	0	0	12.75 ± 0.50 ^Ea^	0	14.87 ± 0.78 ^Fa^	0	17.75 ± 0.70 ^Fa^
0.1% Hb	200 µM Pb	29.25 ± 0.81 ^Bb^	11.52 ± 0.58 ^Fb^	32.47 ± 0.75 ^Bb^	11.92 ± 1.01 ^Fb^	45.37 ± 0.55 ^Bb^	15.12 ± 0.62 ^Fb^
0.1% Hb	200 µM Cd	31.25 ± 0.50 ^Ba^	10.75 ± 0.50 ^Fb^	35.25 ± 0.87 ^Ba^	10.25 ± 0.50 ^Fc^	47.81 ± 0.59 ^Ba^	14.07 ± 0.65 ^Fb^
50 µM L-NAME + 0.1% Hb	0	0	8.67 ± 0.35 ^Fa^	0	10.00 ± 0.60 ^Ga^	0	11.97 ± 0.58 ^Ga^
50 µM L-NAME + 0.1% Hb	200 µM Pb	34.47 ± 0.51 ^Ab^	5.57 ± 0.41 ^Gb^	38.37 ± 0.93 ^Ab^	7.50 ± 0.29 ^Gb^	49.87 ± 0.59 ^Ab^	11.52 ± 0.48 ^Ga^
50 µM L-NAME + 0.1% Hb	200 µM Cd	36.02 ± 0.71 ^Aa^	4.57 ± 0.48 ^Gc^	39.87 ± 0.58 ^Aa^	6.45 ± 0.50 ^Gc^	53.12 ± 0.66 ^Aa^	8.27 ± 0.40 ^Gb^
200 µM SNP	0	0	28.77 ± 0.55 ^Aa^	0	36.77 ± 0.41 ^Aa^	0	51.15 ± 0.90 ^Aa^
200 µM SNP	200 µM Pb	7.62 ± 0.38 ^Hb^	25.45 ± 0.58 ^Ab^	8.50 ± 0.60 ^Hb^	33.75 ± 0.50 ^Ab^	16.55 ± 0.50 ^Gb^	47.55 ± 0.63 ^Ab^
200 µM SNP	200 µM Cd	10.32 ± 0.46 ^Ga^	24.47 ± 0.23 ^Ac^	12.08 ± 0.61 ^Ha^	31.70 ± 0.53 ^Ac^	20.50 ± 1.14 ^Ga^	46.25 ± 1.03 ^Ab^
SNP + L-NAME	0	0	23.50 ± 0.53 ^Ba^	0	29.62 ± 0.45 ^Ba^	0	43.92 ± 0.80 ^Ba^
SNP + L-NAME	200 µM Pb	12.25 ± 0.50 ^Gb^	21.42 ± 0.72 ^Bb^	13.92 ± 0.72 ^Gb^	27.22 ± 0.89 ^Bb^	23.57 ± 0.69 ^Fb^	41.12 ± 0.62 ^Bb^
SNP + L-NAME	200 µM Cd	14.25 ± 0.52 ^Fa^	19.87 ± 0.45 ^Bc^	16.72 ± 0.71 ^Ga^	26.75 ± 0.50 ^Bb^	25.67 ± 0.75 ^Fa^	40.10 ± 0.54 ^Bb^
SNP + Hb	0	0	19.57 ± 0.55 ^Ca^	0	26.02 ± 0.73 ^Ca^	0	38.45 ± 0.38 ^Ca^
SNP + Hb	200 µM Pb	17.27 ± 0.48 ^Fb^	18.05 ± 0.50 ^Cb^	18.60 ± 0.34 ^Fb^	23.60 ± 0.42 ^Cb^	28.55 ± 0.70 ^Eb^	35.45 ± 0.98 ^Cb^
SNP + Hb	200 µM Cd	18.90 ± 0.98 ^Ea^	17.27 ± 0.85 ^Cb^	20.65 ± 1.48 ^Fa^	23.25 ± 0.50 ^Cb^	31.50 ± 0.66 ^Ea^	33.55 ± 0.59 ^Cc^
SNP + L-NAME + Hb	0	0	16.52 ± 0.58 ^Da^	0	22.02 ± 0.58 ^Da^	0	30.72 ± 0.65 ^Da^
SNP + L-NAME + Hb	200 µM Pb	24.25 ± 0.50 ^Db^	15.75 ± 0.50 ^Da^	25.67 ± 0.83 ^Db^	20.15 ± 0.56 ^Db^	33.65 ± 0.77 ^Db^	27.10 ± 0.95 ^Db^
SNP + L-NAME + Hb	200 µM Cd	26.37 ± 0.55 ^Ca^	15.47 ± 0.55 ^Ca^	27.83 ± 1.16 ^Da^	19.05 ± 0.50 ^Dc^	36.17 ± 0.79 ^Da^	26.67 ± 0.48 ^Db^

Each data point is the mean ± SE of four replicates. The treatments contained levels of nitric oxide (L-NAME, N(G)-nitro-L-arginine methyl ester (NO synthase inhibitor)); 0.1% Hb, bovine hemoglobin (NO scavenger), and NO donor, sodium nitroprusside (SNP) alone and in combination with each other, as well as in combination with 200 μM Pb and 200 μM Cd. The capital letters indicated statistically significant differences across various levels of nitric oxide alone or in combination with 200 μM Pb and 200 μM Cd, while the lowercase letters displayed statistically significant differences within each level of nitric oxide alone and in combination with 200 μM Pb and 200 μM Cd, based on Tukey′s test (*p* < 0.05). They are superscripted on top of the numbers.

**Table 3 antioxidants-10-01981-t003:** The impact of various levels of nitric oxide on proline content (Pro), glycine betaine (GB), and glutathione (GSH) content in bamboo species under heavy metals toxicity (200 µM Pb, and 200 µM Cd).

Treament	Proline Content (Pro)	Glycine Betaine (GB)	Glutathione (GSH)
Control	463.75 ± 19.73 ^CDa^	1015.06 ± 23.75 ^Ca^	148.75 ± 3.40 ^Ca^
200 µM Pb	440.0 ± 29.43 ^BCa^	902.18 ± 35.98 ^Db^	141.25 ± 3.40 ^Cab^
200 µM Cd	432.5 ± 28.72 ^BCa^	858.75 ± 39.23 ^Cb^	135.00 ± 4.89 ^Cb^
50 µM L-NAME	367.50 ± 15.00 ^Ea^	712.50 ± 25.00 ^Ea^	115.75 ± 3.86 ^Ea^
50 µM L-NAME + 200 µM Pb	340.0 ± 32.65 ^Dab^	650.00 ± 40.82 ^Fab^	107.50 ± 5.00 ^Eab^
50 µM L-NAME + 200 µM Cd	312.5 ± 22.17 ^DEb^	620.00 ± 37.41 ^Db^	105.25 ± 6.23 ^Db^
0.1% Hb	300.00 ± 24.49 ^Fa^	565.0 ± 40.41 ^Fa^	92.50 ± 5.06 ^Fa^
0.1% Hb + 200 µM Pb	260.0 ± 21.60 ^Eab^	532.50 ± 41.93 ^Gab^	92.50 ± 5.06 ^Fa^
0.1% Hb + 200 µM Cd	242.5 ± 22.17 ^EFb^	490.00 ± 27.08 ^Eb^	83.00 ± 10.23 ^Ea^
50 µM L-NAME + 0.1% Hb	237.50 ± 25.00 ^Ga^	447.50 ± 29.86 ^Ga^	80.25 ± 1.89 ^Ga^
50 µM L-NAME + 0.1% Hb + 200 µM Pb	202.5 ± 22.17 ^Ea^	427.50 ± 43.49 ^Ha^	75.25 ± 3.20 ^Ga^
50 µM L-NAME + 0.1% Hb + 200 µM Cd	200.0 ± 40.82 ^Fa^	387.50 ± 29.86 ^Aa^	66.25 ± 2.50 ^Fb^
200 µM SNP	620.00 ± 40.82 ^Aa^	1407.50 ± 61.84 ^Aa^	211.25 ± 2.62 ^Aa^
200 µM SNP + 200 µM Pb	572.5 ± 28.72 ^Aa^	1320.00 ± 50.99 ^Aab^	188.50 ± 3.00 ^Ab^
200 µM SNP + 200 µM Cd	562.5 ± 25.00 ^Aa^	1260.31 ± 66.22 ^Ab^	185.25 ± 5.43 ^Ab^
SNP + L-NAME	537.50 ± 17.07 ^Ba^	1180.0 ± 28.28 ^Ba^	177.50 ± 5.00 ^Ba^
SNP + L-NAME + 200 µM Pb	500.0 ± 14.14 ^Ba^	1148.75 ± 20.96 ^Ba^	170.75 ± 7.76 ^Ba^
SNP + L-NAME+ 200 µM Cd	500.0 ± 32.65 ^ABa^	1172.50 ± 43.49 ^Ba^	166.50 ± 5.68 ^Ba^
SNP + Hb	485.00 ± 34.39 ^BCa^	1037.50 ± 25.00 ^Ca^	157.50 ± 5.00 ^Ca^
SNP + Hb + 200 µM Pb	480.0 ± 41.63 ^Ba^	993.43 ± 29.11 ^Cab^	149.00 ± 6.05 ^Cab^
SNP + Hb + 200 µM Cd	457.5 ± 28.72 ^Ba^	952.50 ± 41.12 ^Bb^	143.25 ± 6.89 ^Cb^
SNP + L-NAME + Hb	410.00 ± 20.00 ^DEa^	826.25 ± 37.27 ^Da^	136.50 ± 5.06 ^Da^
SNP + L-NAME + Hb + 200 µM Pb	400.0 ± 32.65 ^CDa^	785.62 ± 29.14 ^Ea^	128.00 ± 6.97 ^Dab^
SNP + L-NAME + Hb + 200 µM Cd	380.0 ± 35.59 ^CDa^	712.81 ± 29.40 ^Cb^	119.25 ± 5.50 ^Db^

Each data point is the mean ± SE of four replicates. The treatments contained levels of nitric oxide (L-NAME, N(G)-nitro-L-arginine methyl ester (NO synthase inhibitor); 0.1% Hb, bovine hemoglobin (NO scavenger), and NO donor, sodium nitroprusside (SNP) alone and in combination with each other, as well as in combination with 200 μM Pb and 200 μM Cd. The capital letters indicated statistically significant differences across various levels of nitric oxide alone or in combination with 200 μM Pb and 200 μM Cd, while the lowercase letters displayed statistically significant differences within each level of nitric oxide alone and in combination with 200 μM Pb and 200 μM Cd, based on Tukey′s test (*p* < 0.05). They are superscripted on top of the numbers.

**Table 4 antioxidants-10-01981-t004:** The impact of various levels of nitric oxide on photosynthetic pigments (chlorophylls and carotenoids) in bamboo species under heavy metals toxicity (200 µM Pb and 200 µM Cd).

Treatment	Chl-a	Chl-b	Chl a + b	Caratenoids
Control	10.85 ± 0.74 ^BCa^	8.74 ± 0.58 ^Ca^	20.03 ± 0.76 ^Ca^	1.40 ± 0.31 ^BCa^
200 µM Pb	10.61 ± 0.62 ^BCa^	8.59 ± 0.91 ^Ca^	19.28 ± 0.88 ^Ca^	1.28 ± 0.18 ^Ba^
200 µM Cd	10.32 ± 0.51 ^BCa^	8.22 ± 0.59 ^CDa^	18.78 ± 1.27 ^Ca^	1.36 ± 0.22 ^ABa^
50 µM L-NAME	9.20 ± 0.73 ^DEa^	6.92 ± 0.57 ^DEa^	16.51 ± 0.57 ^DEa^	1.04 ± 0.34 ^Ca^
50 µM L-NAME + 200 µM Pb	9.00 ± 0.66 ^DEa^	7.00 ± 1.09 ^CDa^	15.98 ± 0.77 ^DEa^	1.14 ± 0.70 ^Ba^
50 µM L-NAME + 200 µM Cd	8.58 ± 0.45 ^DEa^	6.77 ± 0.67 ^DEa^	15.55 ± 1.21 ^Da^	1.18 ± 0.39 ^Ba^
0.1% Hb	7.91 ± 0.74 ^EFa^	7.57 ± 0.33 ^CDa^	15.41 ± 1.06 ^Ea^	1.70 ± 0.32 ^ABCa^
0.1% Hb + 200 µM Pb	7.79 ± 0.49 ^EFa^	6.84 ± 0.55 ^DEab^	14.54 ± 0.18 ^Eab^	1.47 ± 0.09 ^Ba^
0.1% Hb + 200 µM Cd	7.58 ± 0.80 ^EFa^	5.83 ± 1.00 ^EFb^	13.45 ± 0.29 ^Eb^	1.17 ± 0.60 ^Ba^
50 µM L-NAME + 0.1% Hb	7.36 ± 0.56 ^Fa^	5.69 ± 0.51 ^Ea^	12.63 ± 1.23 ^Fa^	1.22 ± 0.37 ^Ca^
50 µM L-NAME + 0.1% Hb + 200 µM Pb	7.09 ± 0.69 ^Fa^	5.30 ± 0.77 ^Ea^	12.07 ± 0.69 ^Fab^	1.15 ± 0.17 ^Ba^
50 µM L-NAME + 0.1% Hb + 200 µM Cd	6.72 ± 0.65 ^Fa^	4.47 ± 0.75 ^Ea^	10.90 ± 0.44 ^Fb^	1.05 ± 0.40 ^Ba^
200 µM SNP	12.89 ± 0.39 ^Aa^	12.81 ± 0.73 ^Aa^	25.73 ± 0.80 ^Aa^	2.24 ± 0.27 ^Aa^
200 µM SNP + 200 µM Pb	12.27 ± 0.41 ^Aa^	12.55 ± 0.57 ^Aa^	24.78 ± 1.00 ^Aa^	2.29 ± 0.41 ^Aa^
200 µM SNP + 200 µM Cd	12.18 ± 0.74 ^Aa^	12.16 ± 1.08 ^Aa^	24.28 ± 0.43 ^Aa^	2.13 ± 0.32 ^Aa^
SNP + L-NAME	11.93 ± 0.57 ^ABa^	11.96 ± 0.51 ^ABa^	23.51 ± 0.79 ^Ba^	2.20 ± 0.29 ^Aa^
SNP + L-NAME + 200 µM Pb	11.52 ± 0.38 ^ABa^	10.90 ± 0.52 ^Bab^	22.64 ± 0.97 ^Ba^	1.88 ± 0.30 ^ABa^
SNP + L-NAME + 200 µM Cd	11.46 ± 0.61 ^ABa^	11.22 ± 0.50 ^ABb^	22.32 ± 0.90 ^Ba^	2.06 ± 0.25 ^Aa^
SNP + Hb	11.10 ± 0.54 ^BCa^	10.52 ± 0.57 ^Ba^	21.42 ± 1.34 ^BCa^	1.94 ± 0.10 ^ABa^
SNP + Hb + 200 µM pb	11.01 ± 0.63 ^ABCa^	10.38 ± 0.33 ^Ba^	21.16 ± 1.09 ^BCa^	1.84 ± 0.07 ^ABa^
SNP + Hb + 200 µM Cd	10.69 ± 0.74 ^ABCa^	10.00 ± 1.03 ^BCa^	20.31 ± 0.70 ^Ca^	1.83 ± 0.30 ^ABa^
SNP + L-NAME + Hb	10.21 ± 0.89 ^CDa^	7.86 ± 1.13 ^CDa^	17.77 ± 0.63 ^Da^	1.22 ± 0.33 ^BC^
SNP + L-NAME + Hb + 200 µM Pb	9.84 ± 0.54 ^CDa^	7.46 ± 0.28 ^CDa^	17.08 ± 0.97 ^Da^	1.20 ± 0.14 ^B^
SNP + L-NAME + Hb + 200 µM Cd	9.65 ± 0.66 ^CDa^	7.05 ± 0.81 ^DEa^	16.43 ± 0.45 ^Da^	1.06 ± 0.22 ^B^

Each data point is the mean ± SE of four replicates. The treatments contained levels of nitric oxide (L-NAME, N(G)-nitro-L-arginine methyl ester (NO synthase inhibitor)); 0.1% Hb, bovine hemoglobin (NO scavenger), and NO donor, sodium nitroprusside (SNP) alone and in combination with each other, as well as in combination with 200 μM Pb and 200 μM Cd. The capital letters indicated statistically significant differences across various levels of nitric oxide alone or in combination with 200 μM Pb and 20 μM Cd, while the lowercase letters displayed statistically significant differences within each level of nitric oxide alone and in combination with 200 μM Pb and 200 μM Cd, based on Tukey′s test (*p* < 0.05). They are superscripted on top of the numbers.

**Table 5 antioxidants-10-01981-t005:** The impact of various levels of nitric oxide on translocation factor (TF), bioaccumulation factor (BF) (shoot), and tolerance index (TI) of shoot and root in bamboo species under heavy metals toxicity (200 µM Pb and 200 µM Cd).

Treatment	Translocation Factor (TF)	Tolerance Index (TI) (shoot)	Tolerance Index (TI) (root)	Bioaccumulation Factor (shoot) (BF)
Control	0.00 ± 0.00 ^Cc^	1.00 ± 0.00 ^BCDa^	1.00 ± 0.00 ^CDa^	0.00 ± 0.00 ^Ac^
200 µM Pb	0.64 ± 0.01 ^Ab^	0.95 ± 0.093 ^BCa^	0.95 ± 0.09 ^BCa^	0.10 ± 0.00 ^Db^
200 µM Cd	0.67 ± 0.01 ^Aa^	0.90 ± 0.06 ^BCDa^	0.92 ± 0.01 ^BCa^	0.11 ± 0.00 ^Da^
50 µM L-NAME	0.60 ± 0.01 ^ABb^	0.73 ± 0.07 ^EFa^	0.79 ± 0.01 ^EFa^	0.00 ± 0.00 ^Ac^
50 µM L-NAME + 200 µM Pb	0.64 ± 0.01 ^Aab^	0.69 ± 0.08 ^DEa^	0.75 ± 0.01 ^DEb^	0.12 ± 0.00 ^Cb^
50 µM L-NAME + 200 µM Cd	0.65 ± 0.03 ^Aa^	0.64 ± 0.08 ^EFa^	0.71 ± 0.01 ^DEc^	0.14 ± 0.00 ^Ca^
0.1% Hb	0.71 ± 0.01 ^Aa^	0.87 ± 0.056 ^DEa^	0.69 ± 0.01 ^FGa^	0.00 ± 0.00 ^Ac^
0.1% Hb + 200 µM Pb	0.66 ± 0.02 ^Ab^	0.83 ± 0.09 ^CDa^	0.65 ± 0.07 ^EFa^	0.14 ± 0.00 ^Bb^
0.1% Hb + 200 µM Cd	0.67 ± 0.02 ^Aab^	0.77 ± 0.05 ^DEa^	0.60 ± 0.02 ^Ea^	0.15 ± 0.00 ^Ba^
50 µM L-NAME + 0.1% Hb	0.69 ± 0.11 ^Aa^	0.58 ± 0.07 ^Fa^	0.56 ± 0.09 ^Ga^	0.00 ± 0.00 ^Ac^
50 µM L-NAME + 0.1% Hb + 200 µM Pb	0.63 ± 0.05 ^Aa^	0.53 ± 0.11 ^Ea^	0.50 ± 0.04 ^Fa^	0.16 ± 0.00 ^Ab^
50 µM L-NAME + 0.1% Hb + 200 µM Cd	0.65 ± 0.02 ^Aa^	0.48 ± 0.07 ^Fa^	0.44 ± 0.07 ^Fa^	0.17 ± 0.00 ^Aa^
200 µM SNP	0.56 ± 0.004 ^ABa^	1.30 ± 0.093 ^Aa^	1.26 ± 0.04 ^Aa^	0.00 ± 0.00 ^Ac^
200 µM SNP + 200 µM Pb	0.51 ± 0.01 ^Bb^	1.25 ± 0.11 ^Aa^	1.19 ± 0.009 ^Ab^	0.03 ± 0.00 ^Gb^
200 µM SNP + 200 µM Cd	0.51 ± 0.02 ^Bb^	1.21 ± 0.11 ^Aa^	1.16 ± 0.02 ^Ab^	0.04 ± 0.00 ^Ga^
SNP + L-NAME	0.52 ± 0.03 ^Ba^	1.16 ± 0.082 ^ABa^	1.08 ± 0.05 ^BCa^	0.00 ± 0.00 ^Ac^
SNP + L-NAME + 200 µM Pb	0.51 ± 0.02 ^Ba^	1.14 ± 0.08 ^ABa^	1.06 ± 0.06 ^ABa^	0.06 ± 0.00 ^Fb^
SNP + L-NAME + 200 µM Cd	0.51 ± 0.003 ^Ba^	1.09 ± 0.10 ^ABa^	1.03 ± 0.09 ^ABa^	0.07 ± 0.00 ^Fa^
SNP + Hb	0.50 ± 0.02 ^Bb^	1.06 ± 0.06 ^BCa^	1.15 ± 0.12 ^ABa^	0.00 ± 0.00 ^Ac^
SNP + Hb + 200 µM pb	0.54 ± 0.03 ^Bab^	1.03 ± 0.10 ^ABCa^	1.12 ± 0.08 ^Aa^	0.08 ± 0.00 ^Eb^
SNP + Hb + 200 µM Cd	0.56 ± 0.02 ^Ba^	0.99 ± 0.054 ^BCa^	1.10 ± 0.11 ^Aa^	0.09 ± 0.00 ^Ea^
SNP + L-NAME + Hb	0.61 ± 0.14 ^ABa^	0.90 ± 0.10 ^CDEa^	0.90 ± 0.01 ^DEa^	0.00 ± 0.00 ^Ac^
SNP + L-NAME + Hb + 200 µM Pb	0.65 ± 0.01 ^Aa^	0.879 ± 0.09 ^CDa^	0.86 ± 0.05 ^CDab^	0.12 ± 0.00 ^Cb^
SNP + L-NAME + H b+ 200 µM Cd	0.65 ± 0.02 ^Aa^	0.85 ± 0.12 ^CDa^	0.83 ± 0.01 ^CDb^	0.12 ± 0.00 ^CDa^

Each data point is the mean ± SE of four replicates. The treatments contained levels of nitric oxide (L-NAME, N(G)-nitro-L-arginine methyl ester (NO synthase inhibitor)); 0.1% Hb, bovine hemoglobin (NO scavenger), and NO donor, sodium nitroprusside (SNP) alone and in combination with each other, as well as in combination with 200 μM Pb and 200 μM Cd. The capital letters indicated statistically significant differences across various levels of nitric oxide alone or in combination with 200 μM Pb and 200 μM Cd, while the lowercase letters displayed statistically significant differences within each level of nitric oxide alone and in combination with 200 μM Pb and 200 μM Cd, based on Tukey′s test (*p* < 0.05). They are superscripted on top of the numbers.

## Data Availability

The data presented in this study are available in article.

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
