# Peer review of "Nitric Oxide Ameliorates Plant Metal Toxicity by Increasing Antioxidant Capacity and Reducing Pb and Cd Translocation"

_antioxidants, 2021, doi:10.3390/antiox10121981_

Round 1

Reviewer 1 Report

Title of manuscript too long

In title please provide Cd and Pb instead heavy metals

Abstract

Correct

Key words – correct

Introduction

Please provide aomout - % of area grounds ...

Please add paper about balance macronutrients due to heavy metals

For instance:

Antonkiewicz J., KoÅ‚odziej B., BieliÅ„ska E.J., GleÅ„-Karolczyk K. 2018. The use of macroelements from municipal sewage sludge by the Multiflora rose and the Virginia fanpetals. Journal of Ecological Engineering, 19, 6, 1-13. DOI:  https://doi.org/10.12911/22998993/92889

Material and methods

Were the Certified reference materials (CRM) used in chemical analysis? 

What was standard reference material? 

Please show by methematica formula

Results

Correct

Discussion

Correct

Conclusions

Please add 1-2 sentence about use NO in practice, in phytoextraction of heavy metals by plants from soils .....  

Author Response

Response to Reviewer Comments

Dear Reviewer

Thank you for your valuable comments and thank you for your precious time dedicated to reviewing this paper. The manuscript (antioxidants-1449333) was carefully reviewed, and all the sections and parts were modified according to the comments. Moreover, Linguistically, the manuscript was edited by professor James Barker who is a native speaker of English language. The authors hope that the manuscript in the revised form meets the expectations of the reviewer and can express the content of our research to the community with more clarity.

Best regards

Point 1: Line 1-4

Title of manuscript too long

Response 1: The title was shortened as follow;

Nitric oxide ameliorates plant metal toxicity by increasing antioxidant capacity and reducing Pb and Cd translocation

Point 2: Line 45

Please provide aomout - % of area grounds ...

Response 2:

Cadmium (Cd) is the most toxic heavy metal to humans and the environment and accounts for the highest percentage of heavy metal (7%) toxicity in Chinese agricultural soil and urban areas.

Point 3: Line 48

Please add paper about balance macronutrients due to heavy metals

For instance:

Antonkiewicz J., KoÅ‚odziej B., BieliÅ„ska E.J., GleÅ„-Karolczyk K. 2018. The use of macroelements from municipal sewage sludge by the Multiflora rose and the Virginia fanpetals. Journal of Ecological Engineering, 19, 6, 1-13. DOI:  https://doi.org/10.12911/22998993/92889.

Response 3

The requested correction was applied.( reference number 6)

Point 4: Line 124

Were the Certified reference materials (CRM) used in chemical analysis? 

What was standard reference material?

Response 4:

We used an external standard containing known concentrations of the analytes, which was lake sediment (BCR 280R). We also purchased a known concentration standard of lead and cadmium (bovine liver (SRM 1577c) diluted and analyzed them to give Pb(NO3)2)and (CdCl2)  200 ppm Pb and Cd.  We added these internal standards to a control sample and extracted the lead as we did for our real samples and measured its recovery which was 96% for Pb and 97 % for Cd.

Point 5: Line 231-233

Please show by methematica formula

Response 5:

Translocation factor (TF) = the concentration of NO-pb,Cd in plant shoots (mg/kg)/the concentration of NO- pb, Cd in plant roots.

Tolerance index (TI) of shoot = dry weight of plant shoot from NO- pb, and Cd treatments (g)/ dry weight of plant shoot from control treatment (g).

Tolerance index (TI) of root = dry weight of plant root from NO- pb and Cd treatments (g)/dry weight of plant root from control treatment (g).

TF =

TI(shoot) =

TI(root) =

Point 6: Please add 1-2 sentence about use NO in practice, in phytoextraction of heavy metals by plants from soils ..... 

Response 6: The requested correction was applied. Line 604-608

We concluded that nitric oxide (NO) can play a role in plant perception, signaling, and heavy metals acclimation. Moreover, NO increased the accumulation of the heavy metals in root surface, leading to increased phytoextraction of the heavy metals by the plants from the soil. As a consequence,  it can help to improve the phytoremediation potential of bamboo plants in the environment.

Reviewer 2 Report

NO is an intriguing signaling molecule, as it has both promoting and suppressing effects on cell death, depending on a variety of factors, such as cell type, cellular redox status, and the flux and dose of local NO. This is true also in response to heavy metal toxicity.

For example, in the case of Arabidopsis cell suspension cultures exposed to 100 and 150 mM CdCl2, a process of programmed cell death by accelerating senescence is induced, as reported by De Michele et al., Plant Physiol 2009. In fact, CdCl2 treatment is accompanied by a rapid increase in NO. Hydrogen peroxide production is a later event and precedes the rise of cell death. Inhibition of NO synthesis by L-NMMA, results in partial prevention of H2O2 and mortality, indicating that NO is required for Cd2+ induced cell death. This is only an example, but many papers suggest a role of NO and H2O2 in inducing cell death.

Instead, as shown in this MS, when regenerated Bamboo plants were exposed to 200 mM Cd or Pb in presence of 200 mM SNP as a NO donor, the exogenous application of NO plays a protective role and the plants become more tolerant to the metal toxicity.

In fact, the authors show that in presence of Pb or Cd, the application of NO donors (SNP) in bamboo plants:

  • reduces Pb and Cd accumulation in shoot, stems and roots in plants under heavy metals,
  • improves membrane injury, increases cell protection
  • increases glycine betaine and proline and GSH contents
  • improves antioxidant enzyme activities
  • improves photosynthetic pigments
  • improves plant growth and biomass

The authors suggest that SNPs as NO donors improve the plant defense system in presence of heavy metal (Pb and Cd) toxicity.

Comments

  • the authors do not stress enough the differences in plant response in presence of Cd and Pb when high levels or low levels of NO are present inside the cell. A comparison between high and low NO level effects has to be considered.
  • If I have understood correctly, all these parameters have been measured at the same time (after 30 days treatment). This represents a long time after the beginning of a plant response to the heavy metals exposure. As a consequence, it can only be observed the adaptation of the plant to this stress condition and detected the positive role of high levels of NO that likely contribute to set up rapidly a general plant defense response.

Suggestions:

Experiments at different times after exposures to heavy metals in presence of high levels of NO could help to understand better the effects. Indeed, the positive effects of the exogenous application of high levels of NO have already been reported in many papers.

Last, the authors should improve the figures that result in some aspects unreadable, too small the font size…

In conclusion, I suggest the authors:

  • to enrich the results with experiments performed at different times after heavy metals treatment
  • to complete the introduction and the discussion with the effects of different concentrations of NO in presence of heavy metals
  • to improve the readability of the figures.

Author Response

Response to Reviewer Comments

Dear Reviewer

Thank you for your valuable comments and thank you for your precious time dedicated to  reviewing this paper. The manuscript (antioxidants-1449333) was carefully reviewed, and all the sections and parts were modified according to the comments. Moreover, Linguistically, the manuscript was edited by professor James Barker who is a native speaker of English language. The authors hope that the manuscript in the revised form meets the expectations of the reviewer and can express the content of our research to the community with more clarity.

Best regards

Point 1:

NO is an intriguing signaling molecule, as it has both promoting and suppressing effects on cell death, depending on a variety of factors, such as cell type, cellular redox status, and the flux and dose of local NO. This is true also in response to heavy metal toxicity.

For example, in the case of Arabidopsis cell suspension cultures exposed to 100 and 150 mM CdCl2, a process of programmed cell death by accelerating senescence is induced, as reported by De Michele et al., Plant Physiol 2009. In fact, CdCl2 treatment is accompanied by a rapid increase in NO. Hydrogen peroxide production is a later event and precedes the rise of cell death. Inhibition of NO synthesis by L-NMMA, results in partial prevention of H2O2 and mortality, indicating that NO is required for Cd2+ induced cell death. This is only an example, but many papers suggest a role of NO and H2O2 in inducing cell death.

Instead, as shown in this MS, when regenerated Bamboo plants were exposed to 200 mM Cd or Pb in presence of 200 mM SNP as a NO donor, the exogenous application of NO plays a protective role and the plants become more tolerant to the metal toxicity.

In fact, the authors show that in presence of Pb or Cd, the application of NO donors (SNP) in bamboo plants:

reduces Pb and Cd accumulation in shoot, stems and roots in plants under heavy metals,

improves membrane injury, increases cell protection

increases glycine betaine and proline and GSH contents

improves antioxidant enzyme activities

improves photosynthetic pigments

improves plant growth and biomass

The authors suggest that SNPs as NO donors improve the plant defense system in presence of heavy metal (Pb and Cd) toxicity.

Response 1:

Dear professor Thank you for your comment. That is true. NO has shown contradictory outcomes in the plants subjected to a stressful condition. In particular, the dual impact of NO on the reduction or promoting  heavy metals toxicity and subsequent cell protection or death is intriguing, which seems to be very dependent on the plant type and species, concentrations of heavy metals etc. It could be postulated that NO has non-specific effects on plant cell. Accordingly, as you indicated, there are studies showing that NO encourages cell death (De Michele et al., Plant Physiol 2009). But, on the other hand, a number of studies has reported that NO contributes to preserving plant cells under stressful condition [1,2,3]. In the case of bamboo specious, our result indicated the positive role of NO in scavenging ROS compounds and reducing H2O2 content as well as the stimulation of antioxidant activity and the prevention of translocation of heavy metals from root to aerial plant parts.

  1. Souri Z, Karimi N, Farooq MA, Sandalio LM. Nitric oxide improves tolerance to arsenic stress in Isatis cappadocica desv. Shoots by enhancing antioxidant defenses. Chemosphere. 2020 Jan;239:124523. doi: 10.1016/j.chemosphere.2019.124523. Epub 2019 Aug 12. PMID: 31499308.
  2. Souri Z, Karimi N, Farooq MA, da Silva Lobato AK. Improved physiological defense responses by application of sodium nitroprusside in Isatis cappadocica Desv. under cadmium stress. Physiol Plant. 2021 Sep;173(1):100-115. doi: 10.1111/ppl.13226. Epub 2020 Oct 30. PMID: 33011999.
  3. Jabeen Z, Fayyaz HA, Irshad F, Hussain N, Hassan MN, Li J, et al. (2021) Sodium nitroprusside application improves morphological and physiological attributes of soybean (Glycine max L.) under salinity stress. PLoS ONE 16(4): e0248207. https://doi.org/10.1371/journal.pone.0248207.

Point 2: Line 2

The authors do not stress enough the differences in plant response in presence of Cd and Pb when high levels or low levels of NO are present inside the cell. A comparison between high and low NO level effects has to be considered.

Response 2:

That is true. Such comparison is important. But In in this paper, our aim was to focus on investigating the role of NO in the reduction of heavy metals. Therefore, We were primarily interested in gaining an understanding of whether or not NO mitigates stress in the heavy-metal exposed bamboo plant ? Hence, we used two inhibitors (L-NAME: N(G)-nitro-L-arginine methyl ester (NO synthase inhibitor), Hb: bovine hemoglobin (NO scavenger) and a stimulator (SNP: NO donor)  to make a comparison of performance of the bamboo plant under heavy metals stress with or without NO. This implies that our focus was completely on this area. After obtaining these results from the study,  we can now propose the next question, addressing which level of NO confers the best efficacy in ameliorating toxicity impacts of heavy metals.  

Point 3:

If I have understood correctly, all these parameters have been measured at the same time (after 30 days treatment). This represents a long time after the beginning of a plant response to the heavy metals exposure. As a consequence, it can only be observed the adaptation of the plant to this stress condition and detected the positive role of high levels of NO that likely contribute to set up rapidly a general plant defense response.

Response 3: 

Dear professor, We tried to mimic the environmental condition where the plants in the contaminated areas experiecen a constat exposure to a given abiotic stress including heavy metal, particularly during the early stages of growth and development. 

Point 4:

Suggestions:

Experiments at different times after exposures to heavy metals in presence of high levels of NO could help to understand better the effects. Indeed, the positive effects of the exogenous application of high levels of NO have already been reported in many papers.

Response 4:

Thank you, dear reviewer, for the valuable suggestion. Actually, this is our plan for our next study in which the application of high and low-level levels of NO can help to understand better the effects. The focus of this paper was to pinpoint NO role  in the reduction of heavy metals toxicity with identifying the involved mechanisms. So it can be regarded as a preliminary study before using the high and low levels of NO.

Point 5: Last, the authors should improve the figures that result in some aspects unreadable, too small the font size…

Response 5: The requested correction was applied.

As the esteemed reviewer requested, we reviewed all the figures and reproduced them for increased visibility and readability.

Round 2

Reviewer 2 Report

The authors agreed with my comments, but, if I understand their answers, they do not agree in introducing in the MS any of my suggestions that had the aim to enrich their work.

In fact, as they reported, it is already known that high levels of exogenous NO could increase tolerance to heavy metals in several species.

They try to stress, in this revised version, the potentiality of Bamboo plants for phytoremediation purposes.  In my opinion this is not sufficient to make the information reported in this MS new enough to be published in this Journal.

Author Response

Response to Reviewer Comments

Dear Reviewer

Thank you for your valuable comments and thank you for your precious time dedicated to  reviewing this paper. The manuscript (antioxidants-1449333) was carefully reviewed, and all the sections and parts were modified according to the comments. The authors hope that the manuscript in the revised form meets the expectations of the reviewer and can express the content of our research to the community with more clarity.

Best regards

Comments and Suggestions for Authors

The authors agreed with my comments, but, if I understand their answers, they do not agree in introducing in the MS any of my suggestions that had the aim to enrich their work.

In fact, as they reported, it is already known that high levels of exogenous NO could increase tolerance to heavy metals in several species.

They try to stress, in this revised version, the potentiality of Bamboo plants for phytoremediation purposes.  In my opinion this is not sufficient to make the information reported in this MS new enough to be published in this Journal.

Responses

As the esteemed reviewer requested, we incorporated some new information into the manuscript to enrich the novelty of the results. Hence, we added some complementary reasons and calculated an additional variable (the Bioaccumulation Factor (BF)) to display the functionality of NO in terms of boosting phytoremediation potentiality of the bamboo species.

Complementary reasons

There are many plants that act as hyperaccumulators and can be considered to be used in phytoremediation technology. However, not every hyperaccumulator is a suitable candidate for phytoremediation purposes. In fact, the reports show that very few plants meet all the hyperaccumulators features that are required for phytoremediation. For instance, there are hyperaccumulators with limited root extraction capacities but they may compensate for this shortcoming by having rapid vegetative growth or producing high biomass [1]. Yet such hyperaccumulators are not perfect for phytoremediation. The recent studies have reported that some bamboo species have a high ability to adapt to metalliferous environments and a high capacity to absorb heavy metals [2-3-4]. As a matter of fact, many bamboo plants due to their morphological growth characteristics such as extensive root system and rapid vegetative multiplication as well as high biomass content are suitable for phytoremediation purposes. In particular, the roots, which are the main site of absorption of heavy metal excess by bamboo plants, are capable of transferring the absorbed metals to the aerial parts only when they have already taken up large quantities of heavy metals. Bamboo tissues in the rhizome and culm can accumulate a large number of heavy metals that mainly accumulate in the cell wall, vacuole, and cytoplasm [1]. The potential of species for phytoremediation can be evaluated by two indexes; bioaccumulation factor (BF) and translocation factor (TF) [5-6]. According to the results showin in (Table 2), the heavy metals accumulation value of the bamboo in the roots are higher than stem and shoots indicating that bamboo has a strong ability in accumulation of heavy metals in the root surface .This shows the important role that bamboo plants can play in phytoremediation. This is confirmed by another study [7]. On the other hand, our finding shows that the application of NO can increase the bamboo tolerance by a significant  reduction of bioaccumulation factor (BF) in the shoot as well as by a diminished heavy metal translocation from root to shoot, with an eventual increase in tolerance factor (TF) in the shoot and the root of bamboo species (table 5). Therefore, our results unequivocally demonstrated that NO helps plant phytoremediation in the accumulation of heavy metals on the root surface.

Table 5. The impact of various levels of nitric oxide on translocation factor (TF), bioaccumulation factor (BF) (shoot), and tolerance index (TI) of shoot and root in bamboo species under heavy metals toxicity (200 µM Pb, and 200 µM Cd). Each data point is the mean ± SE of five replicates. The treatments contained four levels of nitric oxide (L-NAME, N(G)-nitro-L-arginine methyl ester (NO synthase inhibitor); 0.1% Hb, bovine hemoglobin (NO scavenger) and, NO donor, sodium nitroprusside (SNP) alone and in combination with each other as well as in combination with 200μM Pb and 200μM Cd. The capital letters indicated statistically significant differences across various levels of nitric oxide alone or in combination with 200μM pb and 200μM Cd, while the lowercase letters displayed statistically significant differences within each level of nitric oxide alone and in combination with 200 μM Pb and 200 μM Cd, based on Tukey′s test(p< 0.05). They are superscripted on top of the numbers .

Treatment

Translocation Factor(TF)

ToleranceIndex (TI) (shoot)

Tolerance Index

(TI)(root)

Bioaccumulation Factor(shoot)(BF)

Control

0.00± 0.00Cc

1.00±0.00BCDa

1.00±0.00CDa

0.00±0.00Ac

200 µM Pb

0.64± 0.01Ab

0.95±0.093BCa

0.95±0.09BCa

0.10±0.00Db

200 µM Cd

0.67± 0.01Aa

0.90±0.06BCDa

0.92±0.01BCa

0.11±0.00Da

50 mM L-NAME

0.60± 0.01ABb

0.73±0.07EFa

0.79±0.01EFa

0.00±0.00Ac

50 mM L-NAME +200 µM Pb

0.64± 0.01Aab

0.69±0.08DEa

0.75±0.01DEb

0.12±0.00Cb

50 mM L-NAME +200 µM Cd

0.65± 0.03Aa

0.64± 0.08EFa

0.71± 0.01DEc

0.14± 0.00Ca

0.1% Hb

0.71± 0.01Aa

0.87±0.056DEa

0.69±0.01FGa

0.00±0.00Ac

0.1% Hb+200 µM Pb

0.66± 0.02Ab

0.83±0.09CDa

0.65±0.07EFa

0.14±0.00Bb

0.1% Hb+200 µM Cd

0.67± 0.02Aab

0.77±0.05DEa

0.60±0.02Ea

0.15±0.00Ba

50 µM L-NAME +.1% Hb

0.69± 0.11Aa

0.58±0.07Fa

0.56±0.09Ga

0.00±0.00Ac

50 µM L-NAME +.1% Hb+200 µM Pb

0.63± 0.05Aa

0.53±0.11Ea

0.50±0.04Fa

0.16±0.00Ab

50 µM L-NAME +.1% Hb+200 µM Cd

0.65± 0.02Aa

0.48±0.07Fa

0.44±0.07Fa

0.17±0.00Aa

200 µM SNP

0.56± 0.004ABa

1.30±0.093Aa

1.26±0.04Aa

0.00±0.00Ac

200 µM SNP+200 µM Pb

0.51± 0.01Bb

1.25± 0.11Aa

1.19± 0.009Ab

0.03± 0.00Gb

200 µM SNP+200 µM Cd

0.51± 0.02Bb

1.21±0.11Aa

1.16±0.02Ab

0.04±0.00Ga

SNP + L-NAME                         

0.52± 0.03Ba

1.16±0.082ABa

1.08±0.05BCa

0.00±0.00Ac

SNP + L-NAME + 200 µM Pb                      

0.51± 0.02Ba

1.14±0.08ABa

1.06±0.06ABa

0.06±0.00Fb

SNP + L-NAME+ 200 µM Cd                      

0.51± 0.003Ba

1.09±0.10ABa

1.03±0.09ABa

0.07±0.00Fa

SNP + Hb

0.50± 0.02Bb

1.06±0.06BCa

1.15±0.12ABa

0.00±0.00Ac

SNP + Hb +200 µM pb                                   

0.54± 0.03Bab

1.03±0.10ABCa

1.12±0.08Aa

0.08±0.00Eb

SNP + Hb+ 200 µM Cd                                  

0.56± 0.02Ba

0.99±0.054BCa

1.10±0.11Aa

0.09±0.00Ea

SNP + L-NAME + Hb

0.61± 0.14ABa

0.90±0.10CDEa

0.90±0.01DEa

0.00±0.00Ac

SNP + L-NAME + Hb+ 200 µM Pb              

0.65± 0.01Aa

0.879± 0.09CDa

0.86±0.05CDab

0.12±0.00Cb

SNP + L-NAME + Hb+ 200 µM Cd              

0.65± 0.02Aa

0.85±0.12CDa

0.83±0.01CDb

0.12±0.00CDa

Dear Reviewer

To our knowledge, this is the first study that evaluates the effect of NO on the reduction of heavy metals toxicity (Pb and Cd) in bamboo plant species to increase bamboo phytoremediation  potential.

  1. Bian, F.; Zhong, Z.; Zhang, X.; Yang, C.; Gai, X. Bamboo – An untapped plant resource for the phytoremediation of heavy metal contaminated soils, Chemosphere, Volume 246, 2020,125750, https://doi.org/10.1016/j.chemosphere.2019.125750.
  2. Wu, L.H.; Liu, Y.J.; Zhou, S.B.; Guo, F.G.; Bi, D.; Guo, X.H.; Baker, A.J.M.; Smith, J.A.C.; Luo, Y.M. Sedum plumbizincicola XH Guo et SB Zhou ex LH Wu (Crassulaceae): a new species from Zhejiang Province, Chin Plant Syst. Evol., 299 (2013), pp. 487-498.
  3. Collin,B.; Doelsch, E.; Keller, C.; Panfili, F.; Meunier.; J.D. Distribution and variability of silicon, copper and zinc in different bamboo species. Plant Soil 2010, 351, 377-387. DOI : 10.1007/s11104-011-0974-9.
  4. Bian, F.Y.; Zhong, Z.K.; Zhang, X.P.; Yang, C.B. Phytoremediation potential of moso bamboo (Phyllostachys pubescens) intercropped with Sedum plumbizincicola in metal-contaminated soil. Environ. Sci. Pollut. Res 2017,24, 27244-27253. doi: 10.1007/s11356-017-0326-2.
  5. Baker, A.J.M.; Brooks, R.R.; Pease, A.J.; Malaisse, F. Studies on copper and cobalt tolerance in three closely related taxa within the genus Silene L.(Caryophyllaceae) from Zaïre. Plant Soil 1983, 73, 377-385. DOI:10.1007/BF02184314.
  6. McGrath, S.P.; Zhao, F.J. Phytoextraction of metals and metalloids from contaminated soils. Curr. Opin. Biotechnol 2003, 14. 277-282. https://doi.org/10.1016/S0958-1669(03)00060-0.
  7. Bian, F.Y.; Zhong, Z.K.; Wu, S.C.; Zhang, X.P.; Yang, C.B.; Xiong, X.Y. Comparison of heavy metal phytoremediation in monoculture and intercropping systems of Phyllostachys praecox and Sedum plumbizincicola in polluted soil. Int. J. Phytoremediation, 2018, 20, 490-498. https://doi.org/10.1080/15226514.2017.1374339.

We provided a response to the  previous comments by the reviewer.

We concur with the reviewer that NO is an intriguing signalling molecule, as it has both promoting and suppressing effects on cell death, depending on a variety of factors including flux and dose of local NO and heavy metal. This is shown in the variety of papers which we have cited.

The Reviewer additionally asks for:

(i)  additional comparison between high and low NO level effects.

(ii) additional measurements at a short time after the beginning of a plant response to the heavy metals exposure.

In relation to these points:

Point (i): There have been several studies using different levels of NO (1-2-3-4). Our paper was however studying this indirectly and more fully by investigating how the activity of NO synthase is affected by NO modulators, such as the application of different l-arginine analogs, including l-NAME (as an NOS inhibitor) and Hb (as an NO scavenger) to demonstrate the role of NO in the enhancement of plant tolerance under abiotic stress.

Point (ii): There have been several studies using different points of reference in relation to time of exposure. At present there in no generally agreed time of exposure to use (quote references ranging from 2 days to 28 days exposure (2-7). The differing kinetics of metal bioavailability in soil under different geological and environmental conditions make the exercise of using controlled time points rather complicated and rarely useful in field. (2-8-9-10)

References

  1. Souri Z, Karimi N, Farooq MA, Sandalio LM. Nitric oxide improves tolerance to arsenic stress in Isatis cappadocica desv. Shoots by enhancing antioxidant defenses. Chemosphere. 2020 Jan;239:124523. doi: 10.1016/j.chemosphere.2019.124523. Epub 2019 Aug 12. PMID: 31499308.
  2. Souri Z, Karimi N, Farooq MA, da Silva Lobato AK. Improved physiological defense responses by application of sodium nitroprusside in Isatis cappadocica Desv. under cadmium stress. Physiol Plant. 2021 Sep;173(1):100-115. doi: 10.1111/ppl.13226. Epub 2020 Oct 30. PMID: 33011999.(14 days)
  3. Jabeen Z, Fayyaz HA, Irshad F, Hussain N, Hassan MN, Li J, et al. (2021) Sodium nitroprusside application improves morphological and physiological attributes of soybean (Glycine max L.) under salinity stress. PLoS ONE 16(4): e0248207. https://doi.org/10.1371/journal.pone.0248207. (28 days)

4- Mostofa, M.G., Seraj, Z.I. & Fujita, M. Exogenous sodium nitroprusside and glutathione alleviate copper toxicity by reducing copper uptake and oxidative damage in rice (Oryza sativa L.) seedlings. Protoplasma 251, 1373–1386 (2014). https://doi.org/10.1007/s00709-014-0639-7(2  days)

5- Liu S, Yang R, Pan Y, Ma M, Pan J, Zhao Y, Cheng Q,Wu M,Wang M, Zhang L (2015b) Nitric oxide contributes to minerals absorption, proton pumps and hormone equilibrium under cadmium excess in Trifolium repens L. plants. Ecotoxicol Environ Safety 119:35–46(7 days).

6- Namdjoyan, S.; Kermanian, H. Exogenous nitric oxide (as sodium nitroprusside) ameliorates arsenic-induced oxidative stress in watercress (Nasturtium officinale R. Br.) plants. Sci. Horticult. 2013, 161, 350–356(7 days).

7-Haitao Shi, Tiantian Ye, Zhulong Chan,Nitric oxide-activated hydrogen sulfide is essential for cadmium stress response in bermudagrass (Cynodon dactylon (L). Pers.),Plant Physiology andBiochemistry,Volume74,2014,Pages99-107,ISSN0981 9428,https://doi.org/10.1016/j.plaphy.2013.11.001.(21 days).

8-Emamverdian, A.; Ding, Y.; Mokhberdoran, F.; Ramakrishnan, M.; Ahmad, Z.; Xie, Y. Different Physiological and Biochemical Responses of Bamboo to the Addition of TiO2 NPs under Heavy Metal Toxicity. Forests 2021, 12, 759. https://doi.org/10.3390/f12060759.

9-Emamverdian, A., Ding, Y., Mokhberdoran, F. et al. Silicon dioxide nanoparticles improve plant growth by enhancing antioxidant enzyme capacity in bamboo (Pleioblastus pygmaeus) under lead toxicity. Trees 34, 469–481 (2020). https://doi.org/10.1007/s00468-019-01929-z.

10- Emamverdian, A.; Ding, Y.; Mokberdoran, F.; Ahmad, Z.; Xie, Y. Determination of heavy metal tolerance threshold in a bamboo species (Arundinaria pygmaea) as treated with silicon dioxide nanoparticles. Glob. Ecol. Conserv. 2020, 24, e0130.

Best Regards,

Round 3

Reviewer 2 Report

The Ms has been enriched with new data and now can be published.